# Mosquito surveillance on U.S military installations as part of a Japanese encephalitis virus detection program: 2016 to 2021

**Mark F. Olson**[1]*, **Caroline Brooks**[1], **Akira Kakazu**[1], **Ploenphit Promma**[2], **Wannapa Sornjai**[2], **Duncan R. Smith**[2], **Timothy J. Davis**[1]

**1** United States Air Force, Pacific Air Forces, Theater Preventive Medicine Flight, Armed Forces Pacific, United States of America, **2** Institute of Molecular Biosciences, Mahidol University, Salaya, Nakhon Pathom, Thailand

* mark.olson.f@gmail.com

**Data Availability Statement:** A CSV file containing data underlying the findings in this study is available in Supporting information.

## Abstract

Japanese encephalitis virus (JEV) continues to circulate throughout Southeast Asia and the Western Pacific where approximately 3 billion people in 24 countries are at risk of infection. Surveillance targeting the mosquito vectors of JEV was conducted at four military installations on Okinawa, Japan, between 2016 and 2021. Out of a total of 10,426 mosquitoes from 20 different species, zero were positive for JEV. The most abundant mosquito species collected were *Aedes albopictus* (36.4%) followed by *Culex sitiens* (24.3%) and *Armigeres subalbatus* (19%). Statistically significant differences in mosquito species populations according to location were observed. Changes in land use over time appear to be correlated with the species and number of mosquitoes trapped in each location. JEV appears to be absent from mosquito populations on Okinawa, but further research on domestic pigs and ardeid birds is warranted.

## Author summary

Japanese encephalitis (JE) is a vector borne disease of significant public health importance for over one third of the earth's population. While JE was first reported in Japan, the most significant outbreaks today are occurring in China and India. Since 2015, Japanese encephalitis vaccine became mandatory for United States military personnel assigned to Japan or South Korea. In an effort to evaluate the risk of JE infection to service members stationed on Okinawa, mosquito surveillance was conducted at four military installations between 2016 and 2021. Over ten thousand mosquitoes were collected, identified to species, and subjected to PCR testing. None of these samples were positive for Japanese encephalitis virus, but further research is still necessary.

**Funding:** This work was financially supported by the Armed Forces Health Surveillance Division—Global Emerging Infections Surveillance (AFHSD-GEIS) under awards P0091_21_18(TD), P0049_22_18(MO), and P0058_23_18(MO). The funders had no role in study design, data collection and analysis, decision to publish, or preparation of the manuscript. No authors received any salary from the funder of this study.

**Competing interests:** We have no conflicts of interest to disclose.

## Introduction

Japanese encephalitis (JE) is among the most devastating viral encephalitides found in the Western Pacific and Southeast Asia, where approximately 3 billion people in 24 countries are at risk of infection [1]. Quan *et al* [2] estimate that in 2015 there were approximately 100,000 cases of JE globally resulting in an estimated 25,000 deaths, with China and India bearing the greatest disease burden.

JE is caused by the Japanese encephalitis virus (JEV), a flavivirus of the family *Flaviviridae* maintained in an enzootic transmission cycle with wading birds as the reservoir, pigs (both wild and domestic) as the amplification host, and mosquitoes as the vector [3]. Domestic pig farms are especially susceptible for viral amplification due to rapid reproduction, erratic animal vaccination, and direct transmission. Serological evidence of JEV in Singapore in the absence of pig farming suggests that other vertebrates may also act as amplification hosts [4]. *Culex tritaeniyorhynchus* Giles, 1901, has historically been considered the primary vector [5,6]. However, JEV has also been detected in 13 additional mosquito species: *Aedes albopictus*, *Ae. vexa*ns, *Ae. vigilax*, *Armigeres subalbatus*, *Cx. annulirostris*, *Cx. bitaeniorhynchus*, *Cx. fuscocephala*, *Cx. gelidus*, *Cx. pipiens*, *Cx. pseudovishnui*, *Cx. quinquefasciatus*, *Cx. sitiens*, and *Cx. vishnui*, all of which were also confirmed as competent vectors of JEV in the laboratory setting [7].

JE was first reported in 1871 in Japan, in both humans and horses [8] with significant outbreaks occurring in Japan, China, India, Guam, Bangladesh, Malaysia and Nepal among other nations between the early 1900s and now [9]. Since then, aggressive vaccination efforts have reduced transmission, preventing an estimated 45,000 infections in 2015 [2]. In 1991, three US military members stationed on Okinawa developed symptoms consistent with encephalitis and subsequently diagnosed with JE and with one patient developing severe neurologic sequelae [10]. None of the three had been vaccinated, nor had any travel off the island prior to onset [10]. Previous to this outbreak, the last known case of JE in Okinawa occurred in 1974 [10]. Since then, there has been 1 case of JE in 1980, one in 1998 and finally, a one-year-old boy with no travel history contracted JE in Okinawa in 2011. In all of Japan, the incidence of JE has been under 10 cases annually since 2017 and there have been zero cases in Okinawa between 2016 and 2022 (Table 1).

The United States Indo-Pacific Command (USINDOPACOM) has approximately 375,000 U.S. military and civilian personnel assigned to this region. Of those, approximately 30,000 are stationed on Okinawa. Since February 1, 2015, Japanese encephalitis vaccine has been mandatory for all active-duty Airmen permanently assigned or temporarily assigned to Japan or

**Table 1. Mosquito-borne disease–Japan, (Okinawa).**

| | Year | | | | | | |
|---|---|---|---|---|---|---|---|
| | **2016** | **2017** | **2018** | **2019** | **2020** | **2021** | **2022** |
| *West Nile Fever* | 0(0) | 0(0) | 0(0) | 0(0) | 0(0) | 0(0) | 0(0) |
| *Yellow Fever* | 0(0) | 0(0) | 0(0) | 0(0) | 0(0) | 0(0) | 0(0) |
| *Zika Fever* | 12(0) | 5(0) | 0(0) | 3(0) | 1(0) | 0(0) | 0(0) |
| *Chikungunya Fever* | 14(0) | 5(0) | 4(0) | 49(0) | 3(1) | 0(0) | 5(0) |
| *Dengue Fever* | 342(4) | 245(2) | 201(0) | 461(10) | 45(0) | 8(0) | 91(3) |
| *Japanese Encephalitis* | 11(0) | 3(0) | 0(0) | 9(0) | 5(0) | 3(0) | 5(0) |
| *Malaria* | 54(0) | 61(1) | 50(1) | 57(0) | 21(0) | 29(0) | 28(0) |

Reference: Official Website of the Okinawa Prefecture (https://www.pref.okinawa.jp/)

South Korea for 30 days or more [11]. The vaccine also became mandatory for members of the US Navy and Marine Corps on November 1, 2016.

In this study, we aim to discover the incidence of JEV among adult mosquitoes in Okinawa between 2016 and 2021. Due to changes in agricultural practices and high rates of vaccination among the Japanese population, we hypothesize JEV circulation to be low.

## Materials and methods

### Study area

Surveillance for JEV was conducted at four military installations on Okinawa from 13 May 2016 through 21 October 2021 (Fig 1). Okinawa (also referred to as the Ryukyu Archipelago) is the southernmost prefecture of Japan, approximately 400 miles south of mainland and is comprised of 150 islands in the East China Sea, the largest of which being the island of Okinawa itself. The climate is subtropical, warm temperate with an average temperature of 22.9° C, the lowest temperature being 17.3° C in January and highest temperature approximately 28.0° C in August, and annual rainfall of 1,817 mm (climate-data.org). The landscape of the

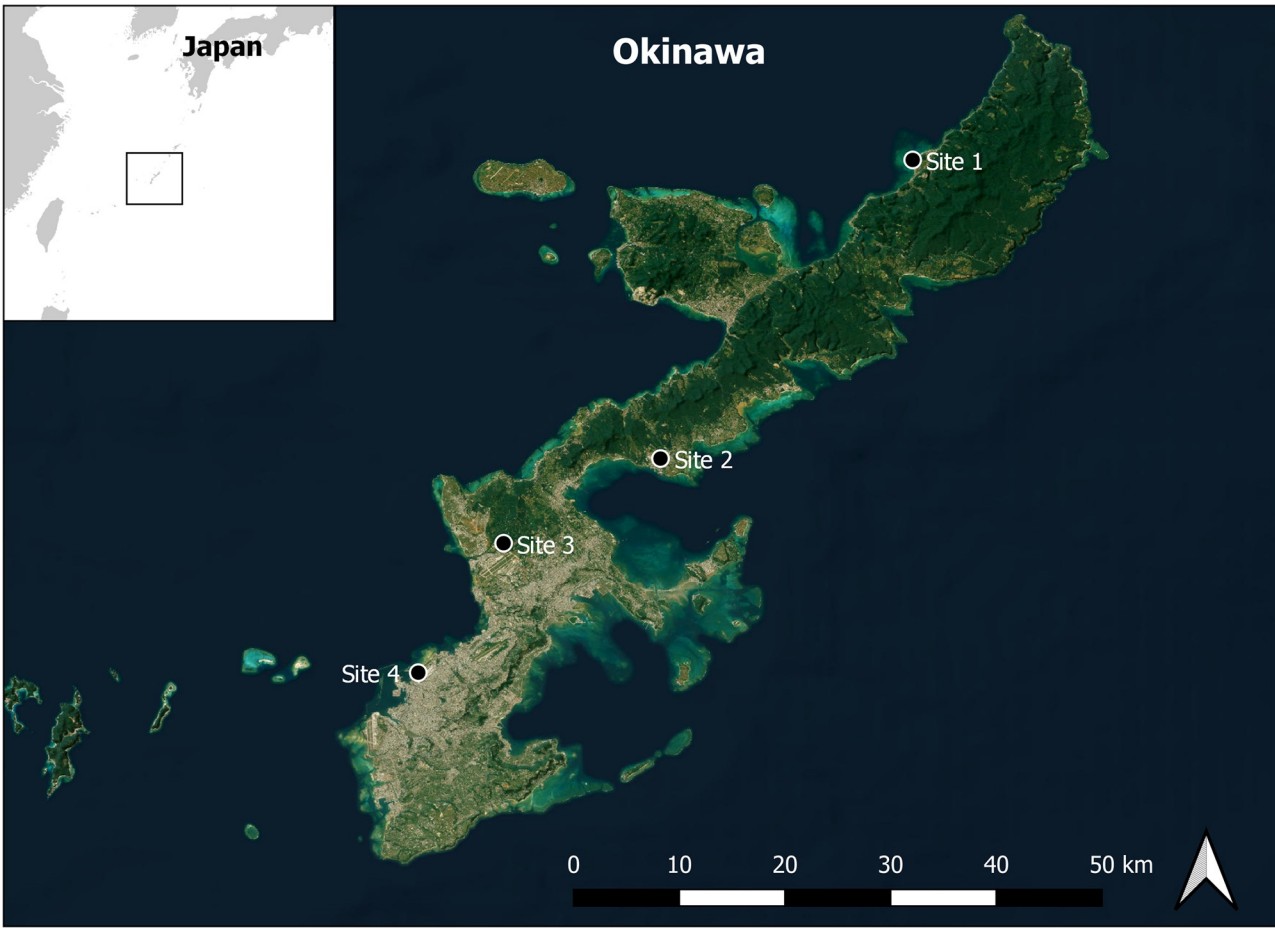

**Fig 1. Step map indicating mosquito trapping locations.** (Map created in QGIS version 3.10.11 A Coruña.) Content is the intellectual property of Esri and is used herein with permission. Copyright 2023 Esri and its licensors. All rights reserved. Site 1: Okuma; Site 2: Hansen; Site 3: Kadena; Site 4: Kinser.

island of Okinawa can roughly be split in half; the northern half more forested and mountainous, comprised mostly of the Yambaru National Forest; and the southern half containing more developed, urban, spaces. Site 1, Okuma Beach Resort (26˚44'17.9" N, 128˚09'32.3" E), is located on a headland in the northern part of the island, surrounded by fields of various crops. Site 2, Camp Hansen (26˚27'22.7" N, 127˚55'14.0" E), is near a small, local farm with goats, chickens and geese. A small river and reservoir are also close to this location, and feral hogs inhabit the forested areas. Site 3 is located on Kadena Air Base on the munitions range (26˚22'28.8" N, 127˚46'13.1" E) which is heavily forested, preserved, and surrounded by various crop fields. Site 4 is at Camp Kinser (26˚15'14.1" N, 127˚41'29.4" E) located near a coastal port and in a densely populated urban part of the island.

## Mosquito sampling

Passive box traps (PBT) (BioQuip Products Inc., Rancho Dominguez, CA) were used to collect mosquito samples using carbon dioxide ($CO_2$) tanks and sugar-baited nucleic acid preservation cards that are red in color and affixed to the left and right side of the inner walls of the trap, as described by Hall-Mendelin et al, Ritchie et al and van den Hurk et al [12–14]. Traps were collected weekly and placed in a large plastic bag and transported in a large cooler to the lab where the entire trap was placed in a freezer to euthanize the adult mosquitoes. Traps at all four locations were replaced on a weekly basis and $CO_2$ flow was monitored and adjusted as needed, and tanks were replaced when empty. Additional sampling was conducted using Mosquito Magnet (MM) (Woodstream Corporation, Lancaster, PA) baited with MM Octenol and a 20-pound propane tank, Reiter/Cummings gravid trap (BioQuip Products Inc., Rancho Dominguez, CA), and Biogents gravid autocidal traps (BG-GAT1) (Biogents, Martinsburg, WV) in 2016 and 2018. A photo of each trap type is available in the supporting information (S1 Fig).

## Identification

Adult mosquitoes were separated by sex and identified morphologically using "The Illustrated Key to Mosquitoes of Okinawa" by the U.S. Army Medical Center Entomology Branch as well as "A revision of the adult and larval mosquitoes of Japan (including the Ryukyu Archipelago and the Ogasawara Islands) and Korea (Diptera: Culicidae)" [15]. Female mosquitoes of the same species were placed into 1.5ml micro centrifuge tubes in pools of no more than 20 mosquitoes per tube. 750 µl of Buffer AVL (Qiagen, Hilden, Germany) was added, and mosquitoes were thoroughly homogenized. Mosquito lysate samples were shipped to Institute of Molecular Biosciences, Mahidol University, Thailand via FedEx for molecular analysis and pathogen detection. Samples were shipped at room temperature with an average transit time of 4 days.

## PCR analysis

RNA was extracted from individual or pooled single species mosquito lysates using QIAamp viral RNA mini-kit (Qiagen, Hilden, Germany) according to the manufacturer's instructions. Complementary DNA was synthesized using RevertAid reverse transcriptase (Thermo Fisher Scientific, Waltham, MA) in a 10 µL reaction containing 4 µL RNA template, 2.5 µM random hexamers (Invitrogen, Carlsbad, CA), 100 units of RevertAid RT, 5 units of RiboLock RNase inhibitor, 1 mM dNTPs and 1X reaction buffer. The thermal cycling conditions were undertaken following manufacturer's protocol.

Pan-flavivirus screening was undertaken using the SuperScript III one-step RT-PCR system (Invitrogen, Carlsbad, CA). The amplification was carried in a 10 µL reaction containing 1 µL of RNA sample, 0.4 µL of Platinum Taq DNA polymerase, 0.4 mM of each dNTP, 3.2 mM

$MgSO_4$ and 200 nM of each pan-flavivirus primers (Flav 100F 5'-AAYTCIACICAIGARATG-TAY-3' and Flav 200R 5'- CCIARCCATRWACCA-3'). The thermal cycling was undertaken according to a previously published protocol [16].

Amplification specifically for JEV was undertaken using DreamTaq DNA polymerase (Thermo Fisher Scientific, Waltham, MA). Reaction was assayed in a 10 μL of PCR mixture containing 1 μL of cDNA, 200 nM dNTPs, 500 μM $MgCl_2$, 0.25 unit of DreamTaq DNA polymerase, 1X DreamTaq buffer and 250 nM of the specific primers (JP1_Fw 5'- GGAAAT-GAAGGCTCAATC-3' and JP2_Rv 5'- GAAGTCACGATTGCCCATTCC-3') according to a previously published protocol [17]. Amplification was undertaken with appropriate negative (no RNA) and positive (stock Beijing-1 JEV RNA) controls. PCR products were separated on 1.0% agarose gels by electrophoresis and products visualized by ethidium bromide staining. Putative positives were rescreened by independent amplification of the lysates and cloning and sequencing of the bands.

We note that the JEV primers used in this study were based on JEV genotype III sequences, and over time this genotype has largely been replaced by genotype I in Asia [18], and as such it is possible that the primers used in the study are less sensitive to detect circulating JEV, although at this point there is no evidence to refute or support this supposition.

### Statistical analysis

To compare the mean count of each mosquito species by collection site and trap type, we used the Wilcoxon rank sum test with continuity correction, adjusting $P$-values ($P \leq 0.05$ is considered significant) with the Benjamini-Hochberg procedure.

We standardized the trap counts to reflect a 7-day trapping period for each trap.

All statistical analyses were conducted in R version 4.2.2 [19] using RStudio version 2022.12.0+353 [20].

## Results

Over the course of this study, a total of 10,426 mosquitoes were captured using all trap methods (Table 2). The total population was evenly distributed among the 4 sites (site 1: n = 2,613; site 2: n = 2,167; site 3: n = 2,501; site 4: n = 3,145). Of these, the most abundant mosquito species was *Ae. albopictus* (n = 3,793; 36.4%), followed by *Cx. sitiens* (n = 2,536; 24.3%), *Ar. subalbatus* (n = 1,985; 19%), and *Cx. quinquefasciatus* (13.7%), respectively. Only 369 (3.5%) *Cx. tritaeniyorhynchus* were trapped over the study period with the highest number being caught in 2019 (n = 59) and zero collected in 2020 and 2021.

Significant differences were noted in trapping method. The MM was only used in 2016 but accounted for 7,215 adult female mosquitoes. The PBT was utilized in all study years and accounted for the next highest number of mosquitoes (n = 3,155), with BG-GAT1 and Reiter/Cummings gravid trap catching a relatively small number (n = 32 and n = 24 respectively). The 7-day mean for the MM was 195.00 (± 43.28 SEM); for the PBT, 30.37 (± 6.48 SEM); for the BG-GAT1, 12.00 (± 6.00 SEM); and for the gravid trap, 44.8 (± 10.52 SEM). The MM and PBT caught significantly higher 7-day mean counts than the gravid trap ($P < 0.005$) and the MM also had significantly higher mean counts than the PBT ($P < 0.0005$).

For analyzing population trends according to geographical location, only mosquitoes caught using the PBT were included (Table 3). The most abundant species trapped at Site 1 was *Ae. albopictus* (n = 1,081; 89.7%). At Site 2, the most abundant species was *Cx. sitiens* (n = 446; 61.3%) but significant numbers of *Ae. albopictus* (n = 98; 13.5%) *Cx. quinquefasciatus* (n = 87;12.0%) and even *Cx. tritaeniyorhynchus* (n = 62; 8.5%) were identified. *Ar. subalbatus* was the dominant species at Site 3 (n = 560; 68.1%) with a significant number of *Ae. albopictus*

**Table 2. Species and number of mosquitoes caught by collection site in Okinawa, Japan, 2016–2021.**

| Species | | | | Site | | Total |
|---|---|---|---|---|---|---|
| | 1 | 2 | 3 | 4 | | |
| *Aedes albopictus* | 2373 | 176 | 475 | 769 | | 3793 |
| *Aedes dorsalis* | | | 2 | | | 2 |
| *Aedes japonicus* | 34 | | 25 | 6 | | 65 |
| *Aedes okinawanus* | 4 | | | | | 4 |
| *Aedes pandani* | | 1 | | | | 1 |
| *Aedes togoi* | | | | 2 | | 2 |
| *Aedes vexans* | 2 | 124 | 2 | 38 | | 166 |
| *Aedes vexans nipponi* | | 14 | | 7 | | 21 |
| *Anopheles sinensis* | 2 | 2 | 1 | | | 5 |
| *Armigeres subalbatus* | 44 | 223 | 1680 | 38 | | 1985 |
| *Coquillettidia crassipes* | | 1 | 4 | | | 5 |
| *Coquillettidia ochrasea* | | 3 | 7 | | | 10 |
| *Culex bitaeniorhynchus* | 2 | 1 | 6 | | | 9 |
| *Culex quinquefasciatus* | 91 | 995 | 253 | 90 | | 1429 |
| *Culex sitiens* | 9 | 450 | 5 | 2072 | | 2536 |
| *Culex tritaeniorhynchus* | 51 | 176 | 21 | 121 | | 369 |
| *Mansonia crassipes* | | | 6 | | | 6 |
| *Mansonia uniformis* | 1 | 1 | | 2 | | 4 |
| *Uranotaenia binoculota* | | | 7 | | | 7 |
| *Uranotaenia lateralis* | | | 7 | | | 7 |
| **Grand Total** | **2613** | **2167** | **2501** | **3145** | | **10,426** |

Site 1: Okuma; Site 2: Hansen; Site 3: Kadena; Site 4: Kinser. Includes passive box trap, Mosquito Magnet, Biogents Gravid Autocidal Trap, and Reiter/Cummings Gravid Trap for surveillance methods.

as well (n = 199;24.2%). Finally, Site 4 is characterized primarily by *Cx. sitiens* (n = 205; 51.1%) and *Ae. albopictus* (n = 119; 29.7%), but also with significant numbers of *Cx. tritaeniyorhynchus* (n = 50; 12.5%).

No mosquitoes or nucleic acid preservation cards tested positive for JEV. Furthermore, none of the samples screened for JEV specifically showed any amplification products after screening with pan-flavivirus primers. However, while a number of flaviviruses including members of the JEV, Kokobera and Nataya virus serocomplexes of the genus *Flavivirus* have the same transmission vectors as JEV [21] to the best of our knowledge none of these viruses have ever been reported in Japan.

A significantly higher mean count per collection of *Ae. albopictus* was observed at Site 1 in relation to site 2–4 ($P < 0.005$). Significantly higher mean populations of *Ar. subalbatus* were found at Site 3 in comparison to Sites 1 ($P = 0.0088$), 2 ($P = 0.0088$) and 4 ($P = 0.0075$). For *Cx. sitiens*, significantly higher mean counts of mosquitoes were observed in Site 2 compared to Site 1 ($P = 0.036$) and Site 3 ($P = 0.036$). The mean count of *Cx. sitiens* captured at Site 4 was also significantly higher than Site 1 ($P = 0.036$). No further statistically significant correlations were observed in other species (*Ae. japonicus, Cx. quinquefasciatus*, and *Cx. tritaenioryhnchus*).

The six most prolific species (having > 40 total mosquitoes as noted in Table 3) found in this study were also analyzed by each site using the previously mentioned statistical analysis (Table 4 and Fig 2). At Site 1, significantly higher mean counts of *Ae. albopictus* than any other

**Table 3. Species and number of mosquitoes caught only with PBT by site in Okinawa, Japan, 2016–2021.**

| Species | | | Site | | Total |
|---|---|---|---|---|---|
| | 1 | 2 | 3 | 4 | |
| *Aedes albopictus* | 1,081 | 98 | 199 | 119 | 1,497 |
| *Aedes dorsalis* | | | 2 | | 2 |
| *Aedes japonicus* | 34 | | 1 | 6 | 41 |
| *Aedes okinawanus* | 4 | | | | 4 |
| *Aedes pandani* | | 1 | | | 1 |
| *Aedes togoi* | | | | 2 | 2 |
| *Aedes vexans* | | 8 | 1 | | 9 |
| *Armigeres subalbatus* | 36 | 20 | 560 | 16 | 632 |
| *Coquillettidia crassipes* | | 1 | 4 | | 5 |
| *Coquillettidia ochrasea* | | 3 | 7 | | 10 |
| *Culex bitaeniorhynchus* | 1 | 1 | 2 | | 4 |
| *Culex quinquefasciatus* | 28 | 87 | 31 | 5 | 151 |
| *Culex sitiens* | 8 | 446 | 5 | 205 | 664 |
| *Culex tritaeniorhynchus* | 12 | 62 | 10 | 50 | 134 |
| *Mansonia uniformis* | 1 | 1 | | 2 | 4 |
| **Grand Total** | **1,205** | **727** | **822** | **401** | **3,155** |

Site 1: Okuma; Site 2: Hansen; Site 3: Kadena; Site 4: Kinser. Includes Passive Box Trap only.

species were observed ($P < 0.005$). At Site 2, *Cx. sitiens* had significantly higher mean counts compared with *Cx. quinquefasciatus* ($P = 0.00083$), *Ar. subalbatus* ($P = 0.01644$) and *Ae. japonicus* ($P < 0.0005$). At Site 3, significantly higher mean counts of *Ar. subalbatus* were detected compared to *Cx. quinquefasciatus*, *Cx. sitiens* and *Cx. tritaenioryhnchus*. However, *Ae. albopictus* populations at this site were also significantly higher than *Ae. japonicus* ($P < 0.005$), *Cx. quinquefasciatus* ($P < 0.005$), and *Cx. tritaeniyorhynchus* ($P < 0.005$). Finally, at Site 4 there were significantly higher mean populations of *Ae. albopictus* compared with *Ae. japonicus* ($P = 0.00058$), *Ar. subalbatus* ($P = 0.00129$), or *Cx. quinquefasciatus* ($P = 0.00129$). Mean counts of *Cx. sitiens* were significantly higher than *Ae. japonicus* ($P = 0.04622$), but not significantly higher than other species.

Significant changes in relative abundance of different mosquito taxa over time were also observed (Fig 3). For example, the mean count of *Ae. albopictus* mosquitoes captured at Site 1 declined from 42.82 (± 17.79 SEM) per trapping event in 2016 to 2.57 (± 0.72 SEM) in 2021.

**Table 4. Mean ± SEM number of female mosquitoes caught per collection (7 trap-nights), by site and species.**

| Species | | Site | | |
|---|---|---|---|---|
| | 1 | 2 | 3 | 4 |
| *Aedes albopictus* | 20.02 ± 4.52 (1,081) | 3.06 ± 0.99 (98) | 6.22 ± 1.98 (199) | 4.71 ± 1.44 (119) |
| *Aedes japonicus* | 0.63 ± 0.34 (34) | - | 0.03 ± 0.03 (1) | 0.27 ± 0.27 (6) |
| *Armigeres subalbatus* | 0.67 ± 0.19 (36) | 0.63 ± 0.33 (20) | 11.81 ± 8.46 (378) | 0.35 ± 0.21 (8) |
| *Culex quinquefasciatus* | 0.52 ± 0.41 (28) | 2.72 ± 2.66 (87) | 0.97 ± 0.87 (31) | 0.22 ± 0.09 (5) |
| *Culex sitiens* | 0.15 ± 0.06 (8) | 13.94 ± 5.07 (446) | 0.16 ± 0.08 (5) | 8.91 ± 4.40 (205) |
| *Culex tritaeniorhynchus* | 0.22 ± 0.09 (12) | 1.94 ± 1.14 (62) | 0.31 ± 0.18 (10) | 2.17 ± 1.16 (50) |

Site 1: Okuma; Site 2: Hansen; Site 3: Kadena; Site 4: Kinser. Includes Passive Box Trap only.

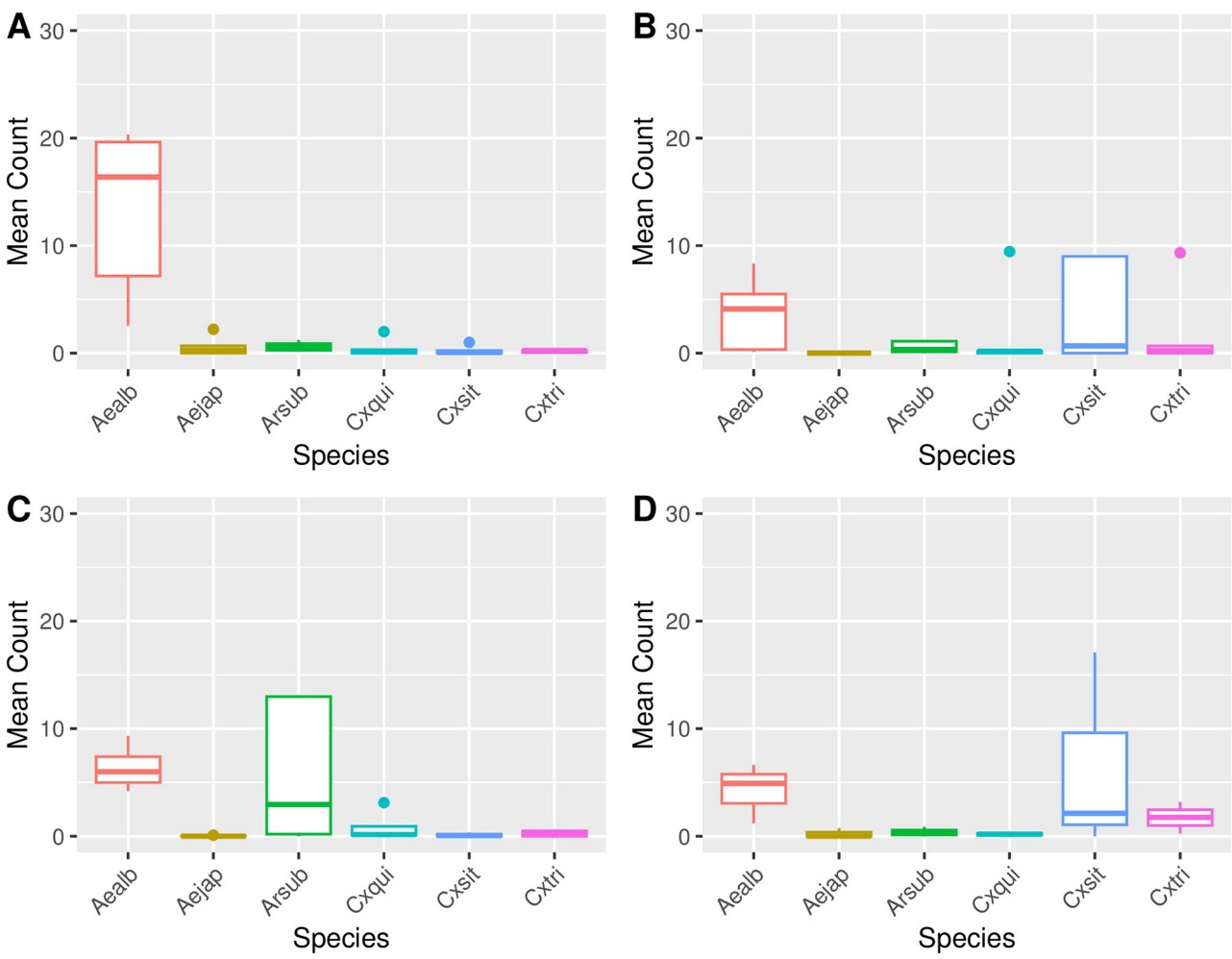

**Fig 2. Mean (± SEM) count of female mosquitoes per trapping event, by species and site, 2016–2021. A**- Site 1, **B**- Site 2, **C**- Site 3, **D**- Site 4. Aealb = *Aedes albopictus*; Aejap = *Aedes japonicus*; Arsub = *Armigeres subalbatus*; Cxqui = *Culex quinquefasciatus*; Cxsit = *Culex sitiens*; Cxtri = *Culex tritaeniyorhynchus*.

Significantly more *Cx. sitiens* were caught in 2021 at Site 2 compared to all previous years ($P <$ .05).

## Discussion

Our study investigated the incidence of JEV in adult mosquitoes captured in Okinawa, Japan, between 2016 and 2021. While several of the confirmed competent vectors of JEV were captured in our surveillance, no JEV was detected. Interestingly, a review of historical data from the Okinawan Prefectural Government showed zero autochthonous cases of Japanese Encephalitis from 2016–2021 (Table 1), and the last recorded military case of JE was in 1991.

The stark absence of Japanese encephalitis virus from adult mosquitoes in Okinawa is likely due to many factors. Changes in agricultural practices have greatly diminished the number of rice paddies on island, a preferred breeding environment for *Cx. tritaeniyorhynchus*. Land use analyses from around 2015, 2019 and 2020 appear to indicate a decrease in "Paddy fields" and

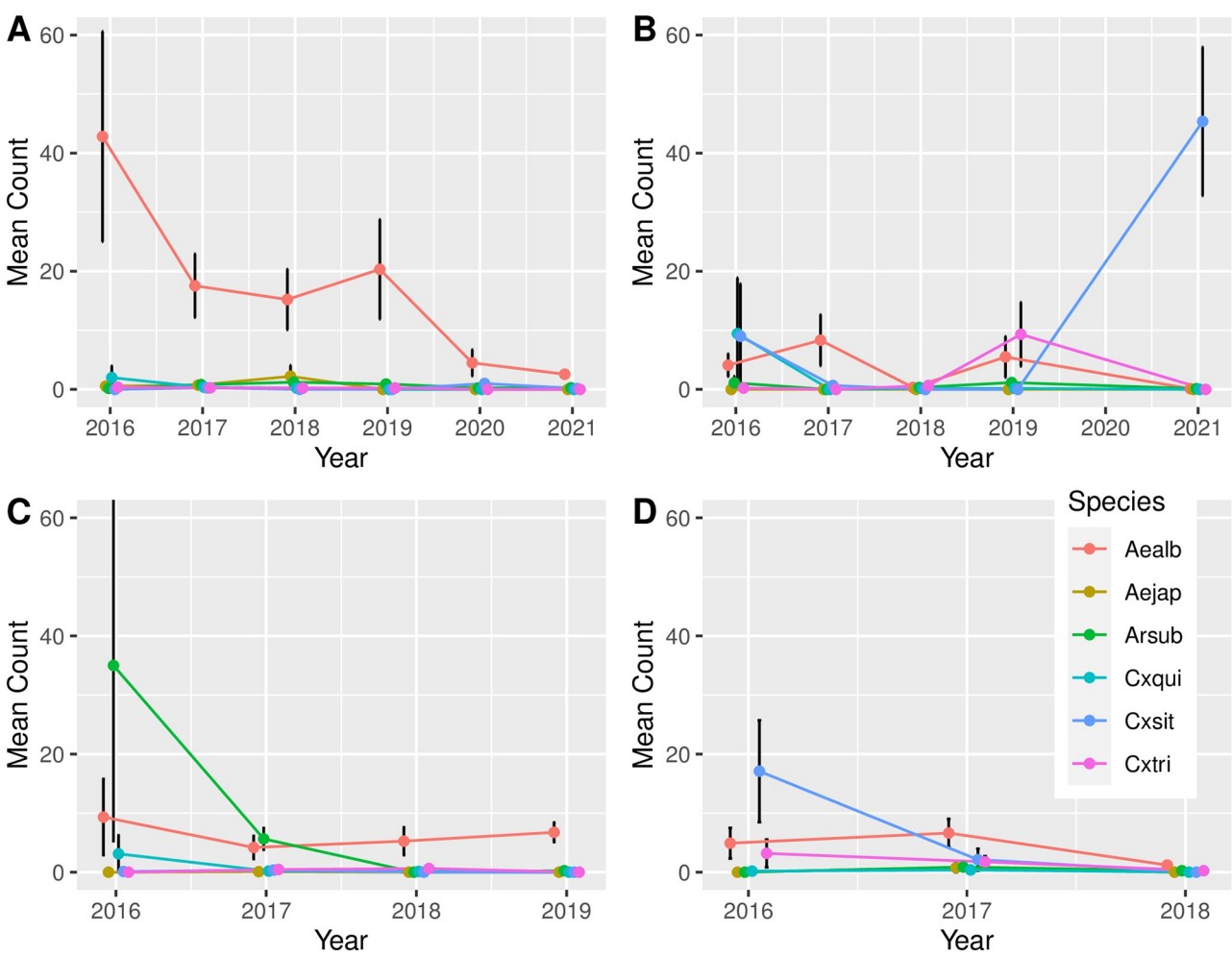

**Fig 3. Change in population relative abundance over time, by species.** Note: Only PBT data was used since PBT was consistently used each year of the study, at all study locations. **A**- Site 1, **B**- Site 2, **C**- Site 3, **D**- Site 4. Aealb = *Aedes albopictus*; Aejap = *Aedes japonicus*; Arsub = *Armigeres subalbatus*; Cxqui = *Culex quinquefasciatus*; Cxsit = *Culex sitiens*; Cxtri = *Culex tritaeniyorhynchus*.

"Crop lands" and an increase in "Built up" areas. (S2 Fig). Post WWII, rice and wheat farming was replaced with sugarcane and other cash crops [22]. These landscape changes, especially the reduction of rice paddies and increases in "Built up" areas likely resulted in diminished mosquito breeding habitats, especially for *Cx. tritaeniyorhynchus* which is also supported by the data we collected.

Pigs and ardeid birds are considered reservoirs and amplifying hosts in the JEV transmission cycle [6] but little is known about the status of JEV in these animals on Okinawa. One study by Nidaira et al [23] found a high percentage of JEV-positive serum samples from wild boars in the Northern part of Okinawa yet they suspected these were infected from JEV that was amplified on pig farms where there was a more constant supply of uninfected hosts. As of 2022, there were 219 pig farms on Okinawa with a total population of 211,700 pigs but we were unable to determine what percentage have been immunized against JEV. Additionally, an unknown population of wild boar is also present on island. Komiya et al [24] found a high rate (86.5%, n = 37) of JEV neutralizing antibodies and 21.6% IgM antibodies (suggesting

recent infection) collected from wild boars in Ishikawa Prefecture, Japan, even during winter months, therefore more research on the status of JEV among these wild animal hosts is warranted.

In our study, the highest number of mosquitoes were collected in 2016. However, this was influenced by the use of the Mosquito Magnet (MM). In 2016, the MM was deployed along with the passive box trap in the same locations for a total of 22 weeks (154 trap nights). During that period, the MM captured 7,215 female mosquitoes compared to 1,661 in the PBT. We discontinued the use of MM after 2016 due to trap malfunction and inability to obtain repair services.

Total mosquito abundance was evenly distributed among trapping sites when considering all trap types. However, heterogeneity of mosquito species diversity and abundance was observed at each collection site when filtering the data for just Passive Box Trap. Site 1, Okuma Beach, seems to favor populations of the Asian tiger mosquito, *Ae. albopictus* and had the greatest overall abundance (Table 3) whereas Site 2, Hansen, appears more suitable to both *Ae. albopictus* and members of the *Cx. sitiens* complex. *Ar. subalbatus* mosquitoes were frequently caught at Site 3, Kadena munitions range, but not at the other locations, and a variety of mosquitoes were caught at Site 4, Kinser, but showed less overall abundance. The greater abundance of *Ae. albopictus* at Site 1 could be a result of its proximity to the evergreen broadleaf forests which comprise the Yambaru National Park. The greatest abundance of *Ar. subalbatus* were captured at Site 3, Kadena AB. The Walter Reed Biosystematics Unit (https://WRBU.si.edu) describes *Ar. subalbatus* as "originally forest-associated" and "thrives in rural and suburban areas." The overall paucity of mosquitoes caught at Site 4 seems to correlate well with the densely urbanized or "built up" conditions potentially offering fewer areas for oviposition.

Currently, vaccination coverage is high in Japan. In 1954, Japan added a mouse brain-derived JEV vaccine to the routine childhood immunization program, but discontinued the recommendation in 2005 due to concerns over potential side effects. Understandably, vaccination coverage dropped dramatically until 2009 when a new, Vero cell-derived vaccine was approved for use and Japan reinstated the recommendation in 2010 [25]. The percentage of the target population of 3–4-year-old children receiving JEV vaccine in 2006 dropped to 4.0%, but climbed to 61.2% in 2009 and in 2020, the Japan Ministry of Health, Labor and Welfare reported 119.1% coverage for the first dose. (https://www.mhlw.go.jp/topics/bcg/other/5.html) (i.e. the target population in 2020 was 939,000 children, but 1,118,107 1st doses of JEV vaccine were administered). Consequently, childhood-onset JE was higher between 2005 and 2015 compared to previous years when the inactivated mouse brain-derived vaccine was in use [25] demonstrating the importance of universal childhood vaccination in locations where JE is still endemic.

Our study had several limitations. In 2020, COVID-19 caused travel restrictions and prevented access to our $CO_2$ vendor. Another limitation in our study was a lack of trap diversity. For the sake of comparison, utilizing the Mosquito Magnet and perhaps the Reiter/Cummings Gravid, Biogents Gravid Autocidal Trap consistently throughout the study would have offered additional insight. Also, homogenized mosquito samples were held at room temperature (24° C) in Buffer AVL for up to 7 days for packaging and shipping to Mahidol University. Recognizing the potential for RNA degradation, our team developed a quality assurance process using positive and negative proficiency analyte testing rounds (blinded to lab personnel) to include with field samples. To date, positive controls undergoing the same packaging and shipping protocol indicate sample integrity is maintained. Finally, we were unable to collect serum samples from domestic pig farms to assess JEV-seropositivity.

In conclusion, we did not detect JEV in any of the mosquitoes we collected between 2016 and 2021. Moreover, with the exception of *Cx. sitiens*, we observed a decrease in the

populations of some of the mosquitoes historically known to serve as vectors for JEV. However, based upon this data alone we cannot determine if JEV has been eliminated from Okinawa. For example, Chen et al [26] demonstrated the potential for JEV transmission by *Ar. subalbatus* in a location without rice farming. Changes in land use (and subsequently mosquito habitat), climate change, and JEV vaccination in humans and animals all play an important role in breaking or facilitating the JEV transmission cycle. Future studies should look at levels of JEV seropositivity in domestic pigs, wild boars as well as herons and other wading birds, and the potential role of *Ar. subalbatus* in the JEV transmission cycle on the island of Okinawa.

## Supporting information

**S1 Fig. Traps used in this study.** A–Mosquito Magnet; B–Passive Box Trap; C–Biogents Gravid Autocidal Trap; D–Reiter/Cummings Gravid Trap. All photos were taken by the author, Dr. Mark F. Olson.
(TIF)

**S2 Fig. Changes in land use at each trapping location over time.** A = Satellite imagery showing trap locations. Content is the intellectual property of Esri and is used herein with permission. Copyright 2023 Esri and its licensors. All rights reserved. B = Land use around 2015. C = Land use around 2019. D = Land use around 2020 (latest version). (Source: ALOS–Advanced Land Observing Satellite, Research and Application Project; https://www.eorc.jaxa.jp/ALOS/en/dataset/lulc_e.htm; Content is the intellectual property of Japan Aerospace Exploration Agency (JAXA) and is used herein with permission).
(TIF)

**S1 Table. CSV file containing data underlying the findings in this study.**
(CSV)

## Acknowledgments

We would like to thank the Okinawa Prefectural Government and University of the Ryukyus for their expertise and collaboration. Additionally, we are grateful to Dr. Gabriel Hamer, Dr. Craig Stoops, and Capt George Cooksey for their review and valuable recommendations for improving the manuscript. We would also like to thank the 18th Wing Public Affairs office at Kadena Air Base for screening this manuscript for operations security (OPSEC) prior to publication.

## Author Contributions

**Conceptualization:** Timothy J. Davis.

**Data curation:** Mark F. Olson, Caroline Brooks, Akira Kakazu, Ploenphit Promma, Wannapa Sornjai, Timothy J. Davis.

**Formal analysis:** Mark F. Olson.

**Funding acquisition:** Mark F. Olson, Timothy J. Davis.

**Investigation:** Mark F. Olson, Akira Kakazu, Ploenphit Promma, Wannapa Sornjai, Duncan R. Smith, Timothy J. Davis.

**Methodology:** Ploenphit Promma, Wannapa Sornjai, Duncan R. Smith, Timothy J. Davis.

**Project administration:** Mark F. Olson, Timothy J. Davis.

**Resources:** Mark F. Olson, Akira Kakazu, Duncan R. Smith.

**Supervision:** Duncan R. Smith.

**Validation:** Duncan R. Smith.

**Visualization:** Mark F. Olson.

**Writing – original draft:** Mark F. Olson, Ploenphit Promma, Wannapa Sornjai.

**Writing – review & editing:** Mark F. Olson, Caroline Brooks, Ploenphit Promma, Wannapa Sornjai, Duncan R. Smith, Timothy J. Davis.

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
