## [Decision Letter · Decision Letter 0]

7 Jul 2023

Dear Dr. Olson,

Thank you very much for submitting your manuscript "Mosquito surveillance on U.S military installations as part of a Japanese encephalitis virus detection program: 2016 to 2021" for consideration at PLOS Neglected Tropical Diseases. As with all papers reviewed by the journal, your manuscript was reviewed by members of the editorial board and by several independent reviewers. The reviewers appreciated the attention to an important topic. Based on the reviews, we are likely to accept this manuscript for publication, providing that you modify the manuscript according to the review recommendations. 

Sincerely,

Mariangela Bonizzoni

Academic Editor

Abdallah Samy

Section Editor

Reviewer's Responses to Questions

**Key Review Criteria Required for Acceptance?**

**Methods**

-Are the objectives of the study clearly articulated with a clear testable hypothesis stated?

-Is the study design appropriate to address the stated objectives?

-Is the population clearly described and appropriate for the hypothesis being tested?

-Is the sample size sufficient to ensure adequate power to address the hypothesis being tested?

-Were correct statistical analysis used to support conclusions?

-Are there concerns about ethical or regulatory requirements being met?

Reviewer #1: Methods are appropriate to test the hypothesis, although more information in some sections would be beneficial to strengthen the data. 

1. Line 95: Are there any mosquito species bias using the passive box traps? Please include the answer/explanation in the text. 

2. Line 101-102: Delete "The sugar-baited nucleic acid preservation cards."

3. Line 114: Please include how the homogenized mosquito samples were shipped. Temperature? Duration of travel? At certain temperatures, short storage periods in buffer AVL could have a significant effect on the viral RNA integrity. 

4. Line 134: Some JEV PCR primers are more sensitive to certain genotypes. Any data that this PCR based on genotype III works well to identify others, including the now dominant genotype Ib? Please include this information in the manuscript.

**Results**

-Does the analysis presented match the analysis plan?

-Are the results clearly and completely presented?

-Are the figures (Tables, Images) of sufficient quality for clarity?

Reviewer #1: The results are very clear and concise. The supplementary figures are especially very visually pleasing and able to highlight the important information right away. 

1. Figure 1: It would be great if sites 1-4 were labeled on the map.

2. Line 175: No positive detection by the pan-flavivirus PCR either? Please include this information. 

3. Line 193: Please include the appropriate p-values for the Site 3 statement.

4. Supplementary figures: Any way to include SFigs 2 and 3 in the main manuscript? They are very helpful as a results summary. Maybe even replace Table 3 or 4 and turn either tables as supplementary tables.

**Conclusions**

-Are the conclusions supported by the data presented?

-Are the limitations of analysis clearly described?

-Do the authors discuss how these data can be helpful to advance our understanding of the topic under study?

-Is public health relevance addressed?

Reviewer #1: Please explain on the study limitations (i.e. any mosquito species bias with the selected mosquito traps).

**Editorial and Data Presentation Modifications?**

Reviewer #1: (No Response)

**Summary and General Comments**

Reviewer #1: The authors submitted a well-written manuscript describing their JEV mosquito surveillance in Okinawa, Japan. Adequate background information was provided in both the introduction and conclusion to highlight the importance of the study and to help the readers digest the data collected. It would be great if the requested additional information is included in the manuscript to strengthen their results and the readers' understanding.

PLOS authors have the option to publish the peer review history of their article (what does this mean?). If published, this will include your full peer review and any attached files.

Reviewer #1: No

Figure Files:

Data Requirements:

Reproducibility:

References

---

## [Editor Report · Decision Letter 1]

5 Oct 2023

Dear Dr. Olson,

We are pleased to inform you that your manuscript 'Mosquito surveillance on U.S military installations as part of a Japanese encephalitis virus detection program: 2016 to 2021' has been provisionally accepted for publication in PLOS Neglected Tropical Diseases.

Best regards,

Mariangela Bonizzoni

Academic Editor

Abdallah Samy

Section Editor

---

## [Editor Report · Acceptance letter]

16 Oct 2023

Dear Dr. Olson,

We are delighted to inform you that your manuscript, "Mosquito surveillance on U.S military installations as part of a Japanese encephalitis virus detection program: 2016 to 2021," has been formally accepted for publication in PLOS Neglected Tropical Diseases.

Best regards,

Shaden Kamhawi

co-Editor-in-Chief

Paul Brindley

co-Editor-in-Chief
